# Enhanced HIV-1 Neutralizing Antibody Breadth in HTLV-2 Co-Infected Individuals: Influence of Antiretroviral Regimen and B Cell Subset Distribution

**DOI:** 10.3390/vaccines13060639

**Published:** 2025-06-13

**Authors:** Eloisa Yuste, María J. Ruiz-De-León, José L. Casado, Ana Moreno, María J. Vivancos, María J. Pérez-Elías, Fernando Dronda, Carmen Quereda, Víctor Sánchez-Merino, Alejandro Vallejo

**Affiliations:** 1National Microbiology Center, Institute of Health Carlos III (ISCIII), 28220 Madrid, Spain; vmsanchez@isciii.es; 2Centro de Investigación Biomédica en Red de Enfermedades Infecciosas (CIBERINFEC), 28222 Madrid, Spainjose.casado@salud.madrid.org (J.L.C.);; 3Laboratory of Immunovirology, Ramón y Cajal Institute of Health Research (IRyCIS), 28034 Madrid, Spain; 4Department of Infectious Diseases, Ramón y Cajal University Hospital, 28034 Madrid, Spain

**Keywords:** HIV-1, neutralizing antibodies, B cell subsets, HTLV-2

## Abstract

Background/Objectives: This study aimed to explore how HTLV-2 infection affects the production of broadly neutralizing antibodies (bNAbs) in persons with HIV-1 (PWH) and to assess the impact of boosted protease inhibitors (PIs). Methods: We evaluated broadly neutralizing antibody (bNAb) activity in 65 PWH, which included 27 who were also co-infected with HTLV-2. All participants were former injection drug users with HCV antibodies and were receiving suppressive antiretroviral therapy (ART). Neutralizing activity was assessed against six recombinant HIV-1 viruses that represent five different subtypes. B cell subsets were also analyzed. Results: HTLV-2 co-infection and the lack of ritonavir-boosted protease inhibitors (r-PIs) were both independently associated with higher neutralization scores (*p* = 0.017 and *p* = 0.005, respectively). Among those not on r-PIs, individuals co-infected with HTLV-2 showed significantly higher neutralization scores (*p* = 0.027) and a broader neutralization breadth (83.4% vs. 48.5%, *p* = 0.015) compared to those infected only with HIV-1. Additionally, HTLV-2 co-infected individuals had more resting memory B cells (*p* = 0.001) and fewer activated memory B cells (*p* = 0.017) than the HIV-1 mono-infected individuals. In our multivariate analysis, only HTLV-2 co-infection remained independently associated with neutralization scores (*p* = 0.027). Elite neutralizers (with a breadth score of ≥10) had more naive B cells and fewer resting memory B cells compared to those with weaker neutralization in both groups. Conclusions: Co-infection with HTLV-2 enhances bNAb production in PWH on suppressive ART and, in particular, in the absence of r-PI regimens. The prominent neutralizing activity corresponded with B cell subset distributions. The results suggest the complexity regarding the interaction between viral co-infections, antiretroviral regimens, and humoral immune compartments and may inform further H1V-1 pathogenesis inquiries or the appropriate design of a vaccine.

## 1. Introduction

HTLV-2 was first discovered in American Indian groups and African Pygmy tribes but quickly spread among intravenous drug users (IDUs), especially in North America and Western Europe [1,2,3,4]. In Spain and other southern European countries, the surge in intravenous drug use during the 1980s significantly contributed to the spread of both HTLV-2 and HIV-1 due to their shared transmission routes [5,6]. Since Spain is not considered endemic for HTLV, blood donations are not mandatorily screened for it, likely leading to underdiagnosed HIV-1/HTLV-2 co-infections. Our hospital serves several penitentiary institutions in Madrid and has noted a high rate of HIV-1/HTLV-2 co-infections, mainly linked to IDU among inmates during the 1980s and 1990s. These co-infections frequently coincided with HCV infections [5,7].

HTLV-2 impacts both HIV-1 and HCV infections. In people with HIV-1 (PWH), HTLV-2 appears to slow CD4^+^ T cell depletion, delay AIDS progression, enhance natural HIV-1 viral load control without antiretroviral therapy, and reduce mortality [8,9,10,11]. Our group demonstrated that CD8+ T cells from co-infected individuals show stronger HIV-1 inhibition in vitro. This is linked to more effector memory cells, higher granzyme and perforin levels, and reduced integrated proviral HIV-1 [12]. Moreover, HTLV-2 may facilitate spontaneous HCV clearance, decrease transaminase levels, and slow fibrosis progression [13], highlighting its potential modulatory role in viral pathogenesis.

While 10–25% of individuals with chronic HIV develop broadly neutralizing antibodies (bNAb), HIV-1 disrupts B cell homeostasis, leading to hypergammaglobulinemia, polyclonal activation, increased turnover, and B cell subset imbalance [14,15,16,17]. Although antiretroviral therapy (ART) can reverse these disruptions [18], ongoing HIV-1 replication perpetuates these abnormalities. However, the role of HTLV-2 in generating broadly neutralizing antibodies (bNAb) or affecting B cell subsets remains unclear. In a recent study, our group analyzed how different antiretroviral treatment regimens influence HIV-1 neutralizing response induction, concluding that ART regimens involving protease inhibitors (PIs) may impact the neutralizing responses [19].

Viral maturation, governed by protease-mediated cleavage of Gag and Gag-Pol precursors, alters envelope spike spatial arrangement. Boosted PIs, combined with agents like ritonavir or cobicistat, enhance drug efficacy, adherence, and resistance barriers. However, our recent data suggest ritonavir-boosted regimens may reduce HIV-1 neutralizing antibody responses, raising important questions about the interplay between ART, viral maturation, and immune responses [20,21,22,23].

This study aimed to explore how HTLV-2 infection affects broadly neutralizing antibody (bNAb) production and assess the impact of boosted protease inhibitors (PIs) in ART regimens for PWH who are also co-infected with HTLV-2. We also investigated possible changes in B cell subsets among co-infected individuals and how these changes might relate to bNAb production.

## 2. Materials and Methods

### 2.1. Study Participants

This research employed a retrospective cross-sectional design examining HIV-1-positive individuals, including those co-infected with HTLV-2, who received care at Madrid’s Ramón y Cajal University Hospital in Spain. All participants had previously used injectable drugs and tested positive for HCV antibodies. Additionally, 12 healthy volunteers served as controls for B lymphocyte subset comparisons. Participants provided written informed consent, and the hospital’s Institutional Ethics Committee approved the study protocol.

### 2.2. Immunoglobulin G Purification and Neutralization Assay

Plasma samples from HIV-positive participants underwent IgG antibody purification using protein A columns (GE Healthcare, Chicago, IL, USA) to eliminate antiretroviral medications that could affect neutralization testing, following established protocols [24,25,26]. The isolated antibodies were evaluated against six engineered replication-capable viruses representing five distinct HIV-1 genetic variants: VI 191 (subtype A, tier 2), 92BR025 (subtype C, tier 1B), 92UG024 (subtype D, tier 2), CM244 (subtype AE, tier 2), NL4-3 (subtype B, tier 1A), and AC10.029 (subtype B, tier 2). An amphotropic vesicular stomatitis virus served as a specificity control.

The experimental procedure involved preparing 96-well plates with virus-free medium (25 μL DMEM with 10% FBS), while purified IgG (0.2 µg/mL in DMEM-FBS) was added in triplicate. Each viral strain was then added, creating 75 μL total volume per well. Viral concentrations were optimized to produce detectable luciferase signals within each strain’s linear response range. Following 1 h incubation at 37 °C, 5 × 10^3^ TZM-bl target cells (BEI Resources, Bethesda, MD, USA, NIAID, NIH: TZM-bl cells, HRP-8129) were added to each well in 100 μL medium. After 72 h incubation in a humidified CO_2_ environment at 37 °C, luciferase activity was quantified using a luminometer and commercial assay kit (Promega, Madison, WI, USA). Neutralization effectiveness was measured in triplicate and expressed as percentage reduction in luciferase activity, indicating decreased viral infectivity. Neutralization percentages were scored as 0 (0–19%, no neutralization), 1 (20–49%), 2 (50–79%), and 3 (>80%). Total neutralization scores were calculated by summing individual virus scores. Cross-subtype neutralization scores reflected the number of viral subtypes showing >50% neutralization activity, with activity against two or more subtypes considered positive cross-neutralization.

### 2.3. Phenotyping of B Lymphocyte Subsets

Cryopreserved PBMCs underwent B lymphocyte phenotyping via multiparameter flow cytometry. Cells were thawed, washed twice in phosphate-buffered saline, and incubated in RPMI medium with 10% fetal bovine serum for 1 h at 37 °C. Surface marker staining utilized the following antibodies: anti-CD3-VioBlue, anti-CD19-FITC, anti-CD27-PE-Vio700, anti-CD10-APC, anti-CD21-PE, and anti-CD20-PerCP-Vio700 (all from Miltenyi Biotech, Bergisch Gladbach, Germany).

Following 1 h antibody incubation at 4 °C, cells were washed twice with PBS containing 0.5% BSA and 0.1% sodium azide, then resuspended in 400 μL PBS-BSA for analysis using MACSQuant 10 instrumentation with MACSQuantify software (version 3.0.1, Bethesda, MD, USA). The gating strategy (Appendix A) first identified single cells, then B lymphocytes (CD3^−^ CD19^+^), which were subdivided based on CD10 and CD27 expression into four primary subsets. Six specific B cell populations were characterized: immature/transitional (CD10^+^ CD27^−^), naive (CD10^−^ CD27^−^ CD21^Hi^), tissue-like memory (CD10^−^ CD27^−^ CD21^lo^), resting memory (CD10^−^ CD27^+^ CD21^Hi^), activated memory (CD10^−^ CD27^+^ CD21^lo^), and plasmablasts (CD10^−^ CD27^++^ CD20^−^ CD21^lo^).

### 2.4. Statistical Analysis

Quantitative data were presented as medians with interquartile ranges (IQ25–75), while categorical data were shown as frequencies and proportions. Independent sample comparisons used the nonparametric Mann–Whitney U test for continuous variables. Correlations between parameters were assessed using Spearman rank correlation analysis. Categorical variable differences were evaluated through contingency table analysis (chi-square distribution). Both univariate and multivariate analyses identified variables independently associated with neutralization scores. All statistical tests were two-tailed with significance set at *p* < 0.05. Analyses were conducted using SPSS Version 22.0 (Chicago, IL, USA), while figures were generated using Prism software version 8.4.3 (GraphPad Software, San Diego, CA, USA).

## 3. Results

This retrospective cross-sectional study included 65 people with HIV-1 (PWH); 27 were co-infected with HTLV-2 (41.5%), and 38 participants had only HIV-1 infection. Table 1 summarizes the demographic and clinical characteristics of all study participants. The two groups showed no significant differences in age, sex, time since HIV-1 diagnosis, duration of antiretroviral therapy (ART), hepatitis C virus infection status, CD4 count, or CD4/CD8 ratio. The healthy control group had a median age of 48 years (IQR, 44–54) and included 25% female participants. Healthy controls showed no significant age or sex differences when compared to the HIV-1 or HIV-1-HTLV-2 co-infected groups (*p* = 0.111 and *p* = 0.098 for age; *p* = 0.358 and *p* = 0.246 for sex).

### 3.1. Impact of Ritonavir-Boosted Regimens

As previously reported by our group [23], ritonavir-boosted protease inhibitor (r-PI) regimens diminish HIV-1 neutralization antibody responses. This was attributed to r-PI interference with viral particle maturation, which hinders the effective neutralizing antibody binding. The PI types in the r-PI regimens were darunavir and lopinavir, with frequencies of 55.5% and 44.4%, respectively, in the co-infected group, and 40% and 60%, respectively, in the HIV-infected group. These regimens had been used for more than three years. In this study, both r-PI presence and HTLV-2 co-infection were independently associated with neutralization scores (univariate analyses, *p* = 0.005 and *p* = 0.017, respectively; Figure 1). No significant associations were found with age, ART duration, time since HIV-1 diagnosis, CD4 nadir, CD4 or CD8 counts, CD4/CD8 ratio, or HCV infection status. However, a marginal association was observed with sex (*p* = 0.075).

### 3.2. HTLV-2 Co-Infection and Neutralization Scores in the Absence of r-PIs

No significant differences in neutralization scores were detected between individuals with or without HTLV-2 co-infection overall (*p* = 0.211; Figure 2A). However, in the absence of r-PIs, the HTLV-2 co-infected group exhibited significantly higher neutralization scores (*p* = 0.027). In individuals on r-PI treatment, observed differences did not reach statistical significance, likely due to neutralizing activity loss attributable to ART. A non-significant trend toward higher neutralization scores was observed in HTLV-2 co-infected males compared to HIV-1-infected males (*p* = 0.072; Figure 2B). HCV infection status did not significantly influence neutralization scores in either group.

The proportion of individuals with cross-reactive neutralizing activity against two or more HIV-1 subtypes was similar between HIV-1 mono-infected (11/33, 33.3%) and HTLV-2 co-infected (9/18, 50%) groups (chi-square test, *p* = 0.281; Figure 2C). However, broader neutralization breadth (scores ≥ 5, encompassing elite, broad, and cross-neutralizing categories) was observed in 48.5% of HIV-1 mono-infected individuals (16/33) compared to 83.4% of HTLV-2 co-infected individuals (15/18, *p* = 0.015, chi-square test; Table 2).

### 3.3. Lymphocyte B Subsets and Neutralizing Activity

Naive B cells were lower in HTLV-2 co-infected individuals compared to HIV-1 mono-infected individuals, though this difference was not statistically significant (*p* = 0.063). Both groups had higher naive B cell counts than healthy controls (*p* < 0.001; Figure 3). Resting memory B cells were significantly higher in HTLV-2 co-infected individuals compared to HIV-1 mono-infected individuals (*p* = 0.001), while activated memory B cells were lower in the HTLV-2 co-infected group (*p* = 0.017). Immature/transitional B cells were slightly higher in HTLV-2 co-infected individuals, though this difference did not reach statistical significance (*p* = 0.066). No significant differences were observed in other B cell subsets, including tissue-like memory B cells and plasmablasts, between the two groups. B cell subsets in the healthy control group were consistent with observations from other studies, considering the age of the individuals [27,28,29].

We further analyzed B cell subsets and HTLV-2 co-infection for their association with neutralization scores. Only HTLV-2 co-infection was independently associated with neutralization scores (linear regression analysis; *p* = 0.027; Figure 4). To explore the relationship between B cell subsets and neutralization capacity further, we stratified individuals based on their neutralization scores (Figure 5). Elite neutralizers (score ≥ 10) demonstrated distinct B cell subset distributions compared to weaker neutralizers (score < 10). In both HTLV-2 co-infected and HIV-1 mono-infected groups, elite neutralizers exhibited significantly higher proportions of naive B cells (*p* = 0.041 and *p* = 0.093, respectively) and lower proportions of resting memory B cells (*p* = 0.019 and *p* = 0.030, respectively). Activated memory B cells were also reduced in elite neutralizers compared to weaker neutralizers, though this difference was more pronounced in the HIV-1 mono-infected group (*p* = 0.099). Notably, when comparing elite neutralizers between groups, HTLV-2 co-infected individuals maintained slightly higher levels of resting memory B cells than HIV-1 mono-infected individuals (*p* = 0.052). This suggests potential HTLV-2-mediated preservation of this B cell subset despite high neutralization activity.

## 4. Discussion

HIV-1 research continues to focus on creating broadly neutralizing antibodies (bNAbs) since they hold promise for infection prevention and control. Antiretroviral therapy (ART) helps restore B cell function despite the humoral immunity impairment caused by HIV-1 infection [18]. However, in individuals co-infected with HTLV-2, the influence of this virus on bNAb response induction is not well characterized. This research examined how HTLV-2 infection affects bNAb creation against six envelope proteins from five HIV-1 subtypes in people with HIV-1 (PWH) who achieved viral suppression through ART.

Ritonavir-boosted protease inhibitor (r-PI) treatment regimens showed decreased neutralization scores, confirming previous research findings. Our data confirms that r-PI treatment might obstruct broadly neutralizing antibody formation, probably due to the disruption of viral maturation processes that generate immature virions unable to be neutralized by antibodies [19,27]. Immature viral particles demonstrate poor efficiency as bNAb targets, which may lead to reduced signals for bNAb production. In fact, protease incorporation into virus-like particles has recently been observed to increase neutralizing responses in immunized mice [30].

HTLV-2 co-infection became an independent determinant that resulted in elevated neutralization scores among PWH, underscoring the complex interactions between multiple viral infections and antibody responses. While other clinical variables—including CD4/CD8 ratios, nadir CD4 counts, and HCV co-infection—did not correlate with neutralization capacity, HTLV-2 co-infection remained a significant determinant. The significant decrease in the CD4/CD8 ratio in co-infected people was due to an increased CD8^+^ T cell count, rather than loss of CD4^+^ T cell function, meaning the helper function necessary for adequate antibody response production appears to be maintained.

All participants in this study were former injecting drug users (IDUs) and had been exposed to HCV. Some achieve sustained virologic response after undergoing HCV treatment. Interestingly, while HCV co-infection did not seem to have a major impact on neutralization capacity, there was a noticeable trend showing higher neutralization scores in males, which warrants further investigation.

Interestingly, among those not on r-PI regimens, HTLV-2 co-infection was linked to significantly broader and more powerful neutralization responses. This suggests that HTLV-2 might actually boost humoral immunity, possibly by influencing immune activation, antigen presentation, or cytokine signaling. Although there was a non-significant difference in cross-subtype neutralization between groups overall, a larger number of individuals co-infected with HTLV-2 fell into the elite or broad neutralizer categories, hinting at a more extensive response in this group.

Those co-infected with HTLV-2 showed higher levels of resting memory B cells and lower levels of activated memory B cells, indicating a shift towards a more balanced memory compartment. This redistribution could be beneficial for long-term maintenance of broadly neutralizing antibody (bNAb) responses and effective affinity maturation. Notably, elite neutralizers (score ≥ 10) in both groups had a higher proportion of naive B cells and a lower proportion of resting memory B cells. This is consistent with previous analyses and may reflect that naive B cells are recruited into active immune responses, while memory B cells are engaged in antibody production and do not accumulate in a resting state [28,29,30]. Of note, elite neutralizers co-infected with HTLV-2 had higher numbers of resting memory B cells compared to those with HIV-1 only, suggesting that HTLV-2 may mediate retention of a responsive memory pool able to respond to various HIV-1 variants.

The relationship between activated memory B cells and neutralization breadth is quite intriguing, as it aligns with earlier research suggesting that ongoing immune activation can hinder high-quality antibody response formation [31,32,33]. Interestingly, HTLV-2 co-infection might actually blunt this immune overactivation while keeping B cell functionality intact, creating a more supportive environment for broadly neutralizing clone emergence. This highlights the importance of balance among B cell subsets—not just frequencies alone—in developing bNAbs.

However, this study has limitations. First, its cross-sectional design does not allow causal inference regarding interactions between HTLV-2 co-infection, B cell activities, and bNAb development. Longitudinal studies are needed to track how immune responses evolve over time. Additionally, the sample size was relatively small, especially among HTLV-2 co-infected individuals not on r-PI treatments, which limits the statistical power and generalizability of these findings. Furthermore, this cohort solely consisted of former IDUs with HCV antibodies, which could pose additional confounding factors. Herpesvirus co-infections (EBV, CMV, and HSV), which can lead to broad immune system activation, were not measured. However, these viruses are detectable in >90% of former IDUs; therefore, we can reasonably assume comparable herpesvirus prevalence between groups.

We also did not consider the molecular characteristics of the antibodies, such as somatic hypermutation, clonal lineages, or epitope specificity, all of which would have provided further insights into the mechanisms. Additionally, we did not assess T follicular helper cells or lymphoid tissue structure, both crucial for antibody maturation. Lastly, we did not include treatment-naive HIV-1 patients as a control group, as this would have introduced confounding variables related to active viral replication, progressive immune dysfunction, and B cell perturbations that occur in untreated HIV infection. Instead, all participants were receiving suppressive ART, providing a controlled baseline where HIV-1 viral loads were undetectable. This approach allowed us to isolate the specific effects of HTLV-2 co-infection on immune responses without the confounding influence of ongoing HIV-1 replication. Previous studies are available that can be applied to B cell subsets for comparative purposes [34,35,36,37,38].

These results align with our previous findings that showed improved CD8^+^ cytotoxic activity in PWH who are co-infected with HTLV-2, contributing to better viral control. Our study provides novel evidence that HTLV-2 can beneficially modulate immune activation by creating a more balanced B cell memory compartment, characterized by increased resting memory B cells and decreased activated memory B cells. This finding is particularly noteworthy given that publications examining humoral immune responses among HTLV-2/HIV-1 co-infected individuals remain scarce in the literature [6,36].

From a clinical perspective, these results suggest the possibility of modifying ART to avoid r-PI regimens in patients co-infected with HTLV-2 to preserve their enhanced humoral responses, which may improve long-term viral control and possibly reduce viral reservoirs. Routine HTLV-2 screening in HIV-positive individuals—especially among those who have previously used injection drugs—could play a crucial role in identifying people with enhanced immune responses who might benefit from tailored treatment approaches. The distinct B cell profiles observed could also act as immunological markers, helping predict bNAb development and inform long-term care strategies.

From a vaccine development perspective, understanding how HTLV-2 modulates immune responses to HIV-1 could lead to new immunotherapy treatments and vaccine designs. The mechanisms involved in HTLV-2-mediated neutralizing antibody production could signal new therapeutic targets for antibody-inducing interventions in patients with isolated HIV infection. Ultimately, these insights underscore the need to consider viral co-infections when managing HIV clinically. The evidence suggests that not every viral co-infection leads to worse outcomes; in fact, some, like HTLV-2, might actually provide immunological advantages that could be utilized therapeutically.

## 5. Conclusions

HTLV-2 co-infection plays a significant role in boosting broadly neutralizing antibody production against HIV-1 in individuals not on ritonavir-boosted protease inhibitors. This phenomenon is linked to a specific B cell subset distribution, showing increased resting memory B cells and decreased activated memory B cells. These findings highlight how viral co-infections and treatment approaches can influence immune responses. Understanding how HTLV-2 affects B cell behavior could help in developing innovative HIV-1 vaccine strategies and immunotherapies that aim to trigger strong and protective bNAb responses.

## Figures and Tables

**Figure 1 vaccines-13-00639-f001:**
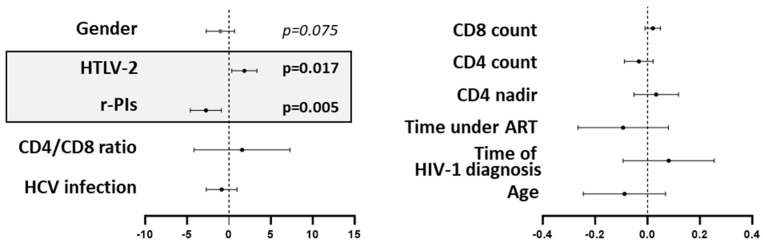
Forest plot illustrating the independent associations of HTLV-2 co-infection and ritonavir-boosted protease inhibitor (r-PI) regimens with HIV-1 neutralizing antibody responses. Effect estimates are presented as adjusted odds ratios (*X*-axis) with 95% confidence intervals from multivariate logistic regression. Statistical significance defined as *p* < 0.05.

**Figure 2 vaccines-13-00639-f002:**
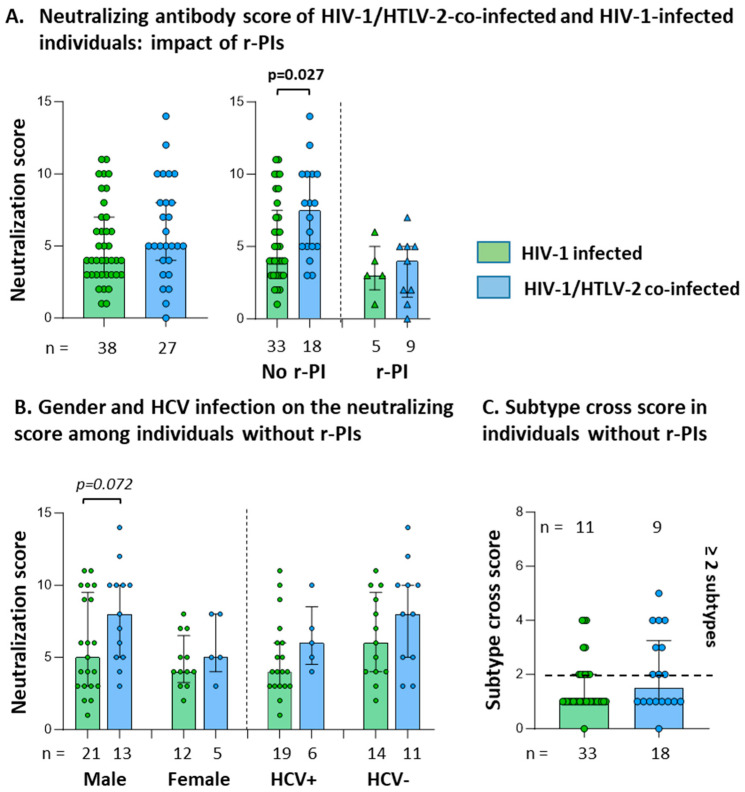
(**A**) Impact of r-PI regimen on the neutralizing antibody score: distribution among all PWH (left panel) and comparison based on r-PI regimen status (right panel) (Mann–Whitney U test). (**B**) Neutralizing antibody scores stratified by sex and HCV co-infection among individuals not receiving an r-PI-based regimen (Mann–Whitney U test). (**C**) Subtype cross-neutralization scores (≥2 subtypes) in individuals not receiving an r-PI regimen (chi-square test). Only comparisons with *p* < 0.1 are shown; statistical significance defined as *p* > 0.05.

**Figure 3 vaccines-13-00639-f003:**
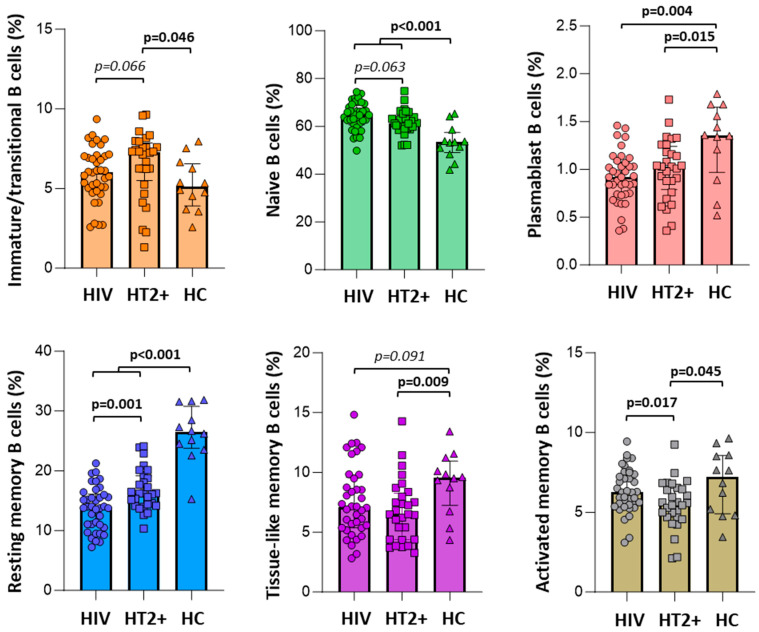
Comparative analysis of B cell subset distributions among HIV mono-infected individuals (HIV), HIV-1-HTLV-2 co-infected individuals (HT2^+^), and healthy controls (HC). Statistical comparisons were performed using the Mann–Whitney U test; only comparisons with *p*-values < 0.1 are shown. Significance threshold set at *p* < 0.05.

**Figure 4 vaccines-13-00639-f004:**
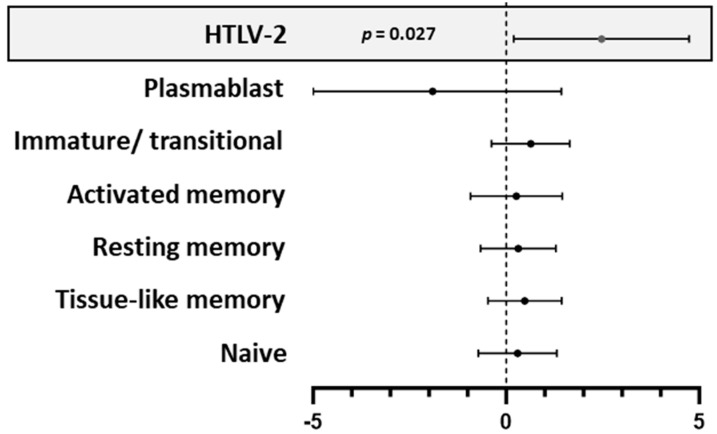
Multivariate linear regression analysis showing the independent association between HTLV-2 co-infection and HIV-1 neutralizing antibody scores. The model was adjusted for potential confounders. Effect estimates are presented as adjusted odds ratios (*X*-axis) with 95% confidence intervals. Statistical significance was defined as *p* < 0.05.

**Figure 5 vaccines-13-00639-f005:**
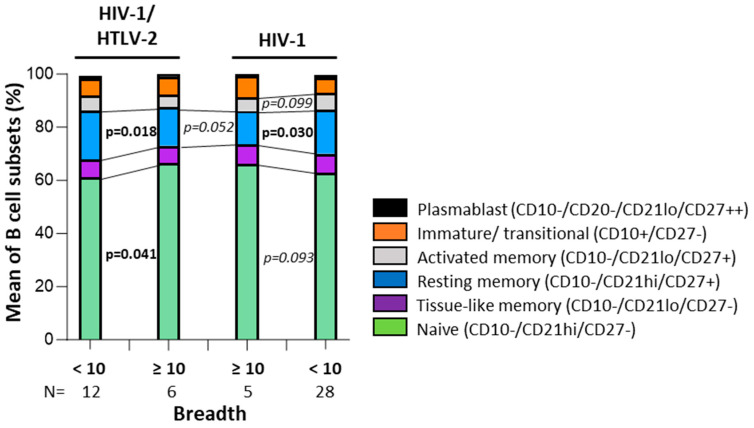
Distribution of B cell subsets (median percentages) stratified by neutralization breadth score in HIV-1 mono-infected and HIV-1-HTLV-2 co-infected individuals. Elite neutralizers (breadth score ≥ 10) exhibited significantly higher frequencies of naive B cells and lower frequencies of resting and activated memory B cells compared to individuals with lower neutralization scores (<10). Comparisons were assessed using the Mann–Whitney U test; only results with *p* < 0.1 are shown. Statistical significance was defined as *p* < 0.05.

**Table 1 vaccines-13-00639-t001:** Demographic and clinical characteristics of HIV-1-infected participants with and without HTLV-2 co-infection.

	HIV-1/HTLV-2 Co-Infected n = 27	HIV-1 Only n = 38	*p* Value
Sex (female, n)	7 (25%)	13 (34%)	0.589
Age (years, median [IQR])	47 [44–52]	48 [45–53]	0.356
Risk practice	IDU (100%)	IDU (100%)	-
HCV antibodies	100%	100%	-
HCV infection, n	9 (36%) *	22 (58%)	0.092
HBsAg (−)/Anti-HBc (+), n HBsAg (+), n	21 (77.7%) 1 (3.7%)	30 (78.9%) 1 (2.6%)	0.781 0.357
Years living with HIV, median [IQR]	24 [20–26]	21 [18–26]	0.292
Years under ART, median [IQR]	17 [13–20]	17 [12–19]	0.760
CD4 nadir (cells/mm^3^), median [IQR]	91 [33–236]	92 [41–138]	0.392
CD4 T cell count (cells/mm^3^), median [IQR]	601 [304–814]	564 [374–833]	0.938
CD8 T cell count (cells/mm^3^), median [IQR]	1068 [803–1510]	870 [637–1151]	0.090
CD4/CD8 ratio, median [IQR]	0.48 [0.33–0.80]	0.67 [0.39–0.89]	0.204
HIV-1 RNA	Undetectable	Undetectable	-
ART for the last three-year period, n 3NRTI 2NRTI + 1 INI 2NRTI + 1 NNRTI 2NRTI + 1 PI 2NRTI + ritonavir-boosted PI	10 (37.1%) 3 (11.1%) 2 (7.4%) 3 (11.1%) 9 (33.3%)	8 (21%) 6 (15.8%) 12 (31.6%) 7 (18.4%) 5 (13.2%)	

IQR, interquartile range; ART, antiretroviral treatment; HBsAg, hepatitis B surface antigen; Anti-HBc, antibody to hepatitis B core antigen; *, three individuals with unknown HCV infection status; IDU, former injecting drug users; NRTI, nucleoside reverse transcriptase inhibitor; NNRTI, non-nucleoside reverse transcriptase inhibitor; INI, integrase inhibitor; PI, protease inhibitor. Mann–Whitney U test and chi-square test. Significant with *p* < 0.05.

**Table 2 vaccines-13-00639-t002:** Distribution of neutralizing antibody breadth categories in HIV-1-infected and HIV-1-HTLV-2 co-infected individuals not receiving ritonavir-boosted protease inhibitors.

Neutralization Score	Breadth Category	HIV-1 Only N = 18	HIV-1/HTLV-2 Co-Infected N = 33
14–18 10–13 5–9 1–4 0	Elite Broad Cross Weak No neutralizer	0 5 (15.1%) 11 (33.3%) 17 (51.5%) 0	1 (5.5%) 5 (27.7%) 9 (50%) 3 (16.6%) 0
	**Combined categories** **(Chi-square test)**		
	Elite + Broad + Cross (≥5)	16 (48.5%)	15 (83.4%)
	Weak + No neutralizers (<5)	17 (51.5%)	3 (16.6%)
			p = 0.015

Significant with *p* < 0.05.

## Data Availability

Data is available at 10.5281/zenodo.15647899.

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
