# Peer review of "Enhanced HIV-1 Neutralizing Antibody Breadth in HTLV-2 Co-Infected Individuals: Influence of Antiretroviral Regimen and B Cell Subset Distribution"

_vaccines, 2025, doi:10.3390/vaccines13060639_

Round 1
Reviewer 1 Report
Comments and Suggestions for Authors
Review of Manuscript vaccines-3655987: "Enhanced HIV-1 Neutralizing Antibody Breadth in HTLV-2 Co-infected Individuals: Influence of Antiretroviral Regimen and B Cell Subset Distribution"
The manuscript offers important information about how HTLV-2 co-infection affects the immune system in people with HIV-1, particularly regarding neutralizing antibody responses and the types of B cell subsets present. Although these findings are pertinent to HIV immunology and vaccine development, the manuscript will be much enhanced by resolving the following problems.
- Authors should clearly describe how these results contribute to the field, particularly regarding earlier research on HTLV-2 and humoral immunity in HIV.
- The authors should talk about how this affects HIV treatment and vaccine creation, as well as how HTLV-2 might help enhance bNAb responses.
- The sample size, especially for subgroup analyses (e.g., by ART regimen or sex), is modest; this may limit the statistical power for some comparisons.
- The text mentions the impact of r-PIs, but it does not include comprehensive analyses of ART regimens that go beyond r-PI use. The study would be strengthened if the authors included more detailed information (such as PI types, time spent on r-PIs, and previous ART history).
- For comparing B cell subsets, healthy controls are included, but it would be interesting to also include HIV-1-infected patients who are not on ART, if possible, for a fuller understanding of the immune system.
- There are minor typographical errors (e.g., “has has demonstrated,” “innates” instead of “inmates,” “progession” instead of “progression”); authors should correct and improve sentence structure for clarity
- Authors should shorten some of the lengthy sentences.
- Authors should ensure that all referenced figures and tables (e.g., Figure 1, Table 1) are clear, appropriately labeled, and support the main finding.
The English could be improved to more clearly express the research
Author Response
Review of Manuscript vaccines-3655987: "Enhanced HIV-1 Neutralizing Antibody Breadth in HTLV-2 Co-infected Individuals: Influence of Antiretroviral Regimen and B Cell Subset Distribution"
The manuscript offers important information about how HTLV-2 co-infection affects the immune system in people with HIV-1, particularly regarding neutralizing antibody responses and the types of B cell subsets present. Although these findings are pertinent to HIV immunology and vaccine development, the manuscript will be much enhanced by resolving the following problems.
Comments 1: Authors should clearly describe how these results contribute to the field, particularly regarding earlier research on HTLV-2 and humoral immunity in HIV.
Response 1: Our study builds upon previous research to make several important contributions to the field. While we confirm our group's earlier findings demonstrating enhanced CD8+ cytotoxic activity in HTLV-2/HIV-1 co-infected individuals, we importantly extend this work by showing that HTLV-2 also enhances humoral immune responses through improved broadly neutralizing antibody production.
This represents a significant advancement, as previous research has primarily focused on the cellular immunity effects of HTLV-2 co-infection. For instance, reference 6 suggests that HTLV-2 modulates the cellular microenvironment to favor its own viability while inhibiting HIV-1 progression through mechanisms including upregulation of viral suppressive chemokines, modulation of the JAK/STAT pathway, and alteration of host miRNA profiles. Similarly, Caterino-de-Araujo et al. (new reference 12 added in the introduction section) demonstrated that individuals co-infected with HIV and HTLV-2 had higher CD4⁺ T cell counts and lower HIV viral loads compared to those with HIV alone or co-infected with HTLV-1, suggesting a protective role of HTLV-2 in HIV infection progression. We added this reference in the text.
We have added this contribution in the Discussion section.
Caterino-de-Araujo A, Campos KR, Oliveira LMS, Rigato PO. Biomarkers in a cohort of HIV-infected patients single- or co-infected with HTLV-1, HTLV-2, and/or HCV: A cross-sectional, observational study. Viruses 2022. https://doi.org/10.3390/v14091955
Comments 2: The authors should talk about how this affects HIV treatment and vaccine creation, as well as how HTLV-2 might help enhance bNAb responses.
Response 2: Our study has significant implications for both HIV treatment and vaccine development strategies.
For treatment, our findings suggest that clinicians should consider avoiding ritonavir-boosted protease inhibitor regimens in HTLV-2 co-infected patients to preserve their enhanced humoral responses, which may improve long-term viral control and potentially reduce viral reservoirs. We recommend implementing routine HTLV-2 screening in HIV-positive individuals, particularly former injection drug users, to identify patients with enhanced immune responses who might benefit from tailored treatment approaches. The distinct B cell profiles observed could serve as immunological biomarkers to predict bNAb development and inform long-term care strategies.
For vaccine development, understanding how HTLV-2 modulates immune responses to HIV-1 could lead to new immunotherapy approaches and vaccine designs. The mechanisms involved in HTLV-2-mediated enhancement of neutralizing antibody production could identify novel therapeutic targets for antibody-inducing interventions in patients with HIV mono-infection, ultimately contributing to the development of more effective HIV prevention and treatment strategies. These strategies warrant further investigation given the scarcity of research on this matter.
These points are addressed in the two last paragraphs of the discussion section.
Comments 3: The sample size, especially for subgroup analyses (e.g., by ART regimen or sex), is modest; this may limit the statistical power for some comparisons.
Response 3: We noted this limitation in the discussion, stating that the sample size was relatively small, especially among the subgroup of HTLV-2 co-infected individuals not on r-PI treatments, which limits statistical power and generalizability of findings. These limitations suggest that larger, multi-center studies would be needed to confirm these preliminary findings and provide more robust evidence for clinical decision-making.
The HTLV-2/HIV co-infected population remains limited in size, and new infections are increasingly uncommon due to evolving practices among individuals with a history of intravenous drug use.
Comments 4: The text mentions the impact of r-PIs, but it does not include comprehensive analyses of ART regimens that go beyond r-PI use. The study would be strengthened if the authors included more detailed information (such as PI types, time spent on r-PIs, and previous ART history).
Response 4: We appreciate the comment and agree that a more detailed analysis of the different ART regimens would be of interest for the present study. However, the sample size limits the feasibility of such an analysis in this case. In contrast, a recent study published by our group included a cohort of 101 PLWH with ART-suppressed viremia, which allowed for a more comprehensive analysis considering different treatment regimens and other immunological factors (https://doi.org/10.3390/vaccines12101176). The conclusion of that study was similar to the present one: in both studies, treatment with b-PIs was associated with reduced neutralization.
To clarify this point, we have revised the introduction section in the revised version.
Nevertheless, PI types and time under r-PIs are detailed in the Results section. Previous ART history presents a more complex challenge to document comprehensively, as most participants in this study have been receiving different ART regimens for over 17 years. Therefore, Table 1 includes the ART regimens used by both groups over the most recent three-year period.
Comments 5:For comparing B cell subsets, healthy controls are included, but it would be interesting to also include HIV-1-infected patients who are not on ART, if possible, for a fuller understanding of the immune system.
Response 5: Although the inclusion of HIV-1-infected patients not receiving antiretroviral therapy would provide a more comprehensive understanding of immune system changes, the absence of this control group in the present study can be justified by several considerations:
- Including untreated patients would have introduced confounding variables related to active viral replication, progressive immune dysfunction, and B-cell perturbations that occur in untreated HIV infection.
- All participants were receiving suppressive antiretroviral therapy (ART), which provided a controlled baseline with undetectable HIV-1 viral loads. This approach allowed the researchers to isolate the specific effects of HTLV-2 coinfection on immune responses without the confounding influence of ongoing HIV-1 replication.
We have acknowledged this limitation in the discussion section and included additional references for comparative purposes:
Buckner CM, Moir S, Ho J, et al. Characterization of plasmablasts in the blood of HIV-infected viremic individuals: evidence for nonspecific immune activation. J Virol. 2013; 87: 5800-5811. DOI: 10.1128/JVI.00094-13
Morbach H, Eichhorn EM, Liese JG, Girschick HJ. Reference values for B cell subpopulations from infancy to adulthood. Clin Exp Immunol 2010; 162: 271-279. DOI: 10.1111/j.1365-2249.2010.04206.x.
Amu S, Lavy-Shahaf G, Cagigi A, et al. Frequency and phenotype of B cell subpopulations in young and aged HIV-1 infected patients receiving ART. Retrovirology 2014; 11: 76. DOI: 10.1186/s12977-014-0076-x
Moir S, Buckner CM, Ho J, et al. B cells in early and chronic HIV infection: evidence for preservation of immune function associated with early initiation of antiretroviral therapy. Blood 2010; 116: 5571-5579. DOI: 10.1182/blood-2010-05-285528
Comments 6: There are minor typographical errors (e.g., “has has demonstrated,” “innates” instead of “inmates,” “progession” instead of “progression”); authors should correct and improve sentence structure for clarity
Response 6: Proper proofreading and editorial review have been performed
Comments 7: Authors should shorten some of the lengthy sentences.
Response 7: It has been corrected
Comments 8: Authors should ensure that all referenced figures and tables (e.g., Figure 1, Table 1) are clear, appropriately labeled, and support the main finding.
Response 8: It has been revised
Reviewer 2 Report
Comments and Suggestions for Authors
This is a highly intriguing and well-documented report on the effect of HTLV-2 co-infection on the magnitude and breadth of HIV-1 neutralizing antibodies. The lab performs actual neutralizing assays when many groups send out plasma or serum samples for psudotype analysis. The analysis and statistics for antibody assays and flow cytometry are straightforward and appropriate. Some typos were noted: a) The N in Table 2 for HIV only is incorrect, b)Line 314: blunted should be blunt. A number of questions come to mind which the authors might address in the text:
- It is interesting that they isolate IgG when most labs use serum or plasma. Was there a reason for that?
- Was the HTLV-2 effect virus specific? Do the co-infected individuals also have higher levels of antibody to CMV or HSV, or EBV (which should be detectable in >90% of former IV drug users)?
- Is there any antigenic cross reactivity between HTLV-2 and HIV-1?
- Is HTLV-2 known to have any protein mimicking a human cytokine that modulates inflammation (CMV has an IL-10 like protein)?
- Have any assays been conducted to examine monocyte expression of TNF or IL-6 in the HTLV-2 co-infected subjects relative to the monoinfected?
- Were subjects tested for HBV antibodies?
Author Response
This is a highly intriguing and well-documented report on the effect of HTLV-2 co-infection on the magnitude and breadth of HIV-1 neutralizing antibodies. The lab performs actual neutralizing assays when many groups send out plasma or serum samples for psudotype analysis. The analysis and statistics for antibody assays and flow cytometry are straightforward and appropriate. Some typos were noted: a) The N in Table 2 for HIV only is incorrect, b) Line 314: blunted should be blunt. A number of questions come to mind which the authors might address in the text:
Comments 1: It is interesting that they isolate IgG when most labs use serum or plasma. Was there a reason for that?
Response 1: It has been previously reported that residual antiretroviral compounds present in the serum of individuals undergoing ART can inhibit HIV replication, in some cases with considerable potency, potentially confounding the interpretation of neutralization assays. To address this, and in order to specifically assess the contribution of HIV-specific antibodies to viral neutralization, we isolated serum IgG and evaluated its neutralizing activity against a panel of representative viruses. This approach, which has also been adopted by other groups in similar contexts, allows for a more accurate characterization of humoral immune responses in ART-treated individuals. These studies are included as new references:
Esmaeilzadeh E, Etemad B, Lavine CL, Garneau L, Li Y, Regan J, Wong C, Sharaf R, Connick E, Volberding P, Sagar M, Seaman MS, Li JZ. Autologous neutralizing antibodies increase with early antiretroviral therapy and shape HIV rebound after treatment interruption. Sci Transl Med 2023; 15:eabq4490. doi: 10.1126/scitranslmed.abq4490.
Bertagnolli LN, Varriale J, Sweet S, Brockhurst J, Simonetti FR, White J, Beg S, Lynn K, Mounzer K, Frank I, Tebas P, Bar KJ, Montaner LJ, Siliciano RF, Siliciano JD. Autologous IgG antibodies block outgrowth of a substantial but variable fraction of viruses in the latent reservoir for HIV-1. Proc Natl Acad Sci U S A 2020; 117:32066-32077. doi: 10.1073/pnas.2020617117
We appreciate the reviewer’s comment and have modified the corresponding Material and Methods section to provide further clarification.
Comments 2: Was the HTLV-2 effect virus specific? Do the co-infected individuals also have higher levels of antibody to CMV or HSV, or EBV (which should be detectable in >90% of former IV drug users)?
Response 2: We did not assess antibody levels against herpesviruses such as CMV, HSV, or EBV. However, as you noted, these viruses are detectable in >90% of former intravenous drug users. Given that both study groups consisted of former injecting drug users, we can reasonably assume comparable herpesvirus prevalence between groups.
Comments 3: Is there any antigenic cross reactivity between HTLV-2 and HIV-1?
Response 3: Both are retroviruses but belong to different genera (HIV-1 is a lentivirus, HTLV-2 is a deltaretrovirus). They have different envelope proteins and tropisms. Significant antigenic cross-reactivity would be unexpected given their distinct phylogenetic origins. Nevertheless, limited serological cross-reactivity has been reported that can lead to indeterminate Western blot results, particularly in certain populations. While this may complicate diagnostic interpretation, it does not represent clinically significant cross-reactivity.
Comments 4: Is HTLV-2 known to have any protein mimicking a human cytokine that modulates inflammation (CMV has an IL-10 like protein)?
Response 4: HTLV-2 is not known to encode a protein that directly mimics human cytokines like the IL-10 homolog produced by cytomegalovirus (CMV). However, HTLV-2 can modulate the host immune response through indirect mechanisms, such as altering cytokine production or signaling pathways in infected cells.
For example: HTLV-2 encodes the Tax-2 protein, which can activate NF-κB and other transcription factors, leading to the upregulation of pro-inflammatory cytokines (e.g., IL-6, TNF-α) and anti-inflammatory cytokines (e.g., IL-10) in infected T cells. While Tax-2 does not mimic a human cytokine, it dysregulates cytokine networks; and HTLV-2 infection has been associated with reduced immune activation in co-infected HIV-1 individuals, possibly through altered cytokine profiles (e.g., increased IL-10 or IFN-γ) or effects on T-cell and B-cell homeostasis. HTLV-2 also upregulates the expression of MIP-1α, MIP-1β, RANTES that are co ligand for the CCR5 coreceptor, blocking the interaction of HIV with the CD4 cell.
Comments 5: Have any assays been conducted to examine monocyte expression of TNF or IL-6 in the HTLV-2 co-infected subjects relative to the monoinfected?
Response 5: This study does not examine the expression of TNF or IL6 but we extensively examined these citokines in other published studies from our group where HTLV-2 coinfected individuals showed higher levels of both cytokines compared to hiv monoinfected individals. These results are reported in Viruses 2022 (ref 12) and AIDS 2015 (ref 13). This confirm the role of Tax-2 protein in the upregulation of these pro-inflammatory cytokines.
Comments 6: Were subjects tested for HBV antibodies?
Response 6: HBV infection status has been included in Table 1. The majority of patients showed evidence of past HBV infection, with no significant differences between groups.
Reviewer 3 Report
Comments and Suggestions for Authors
This is very interesting and well conducted study, and a well written manuscript. The authors have investigated the impact of HTLV-2 on the presence of broadly neutralizing anti-HIV antibodies in HIV-infected individuals. They also looked into their effects of B cell subsets. It is sad that the researchers could not look into the frequencies of circulating TFH in the study participants. Nevertheless, the study provides very novel insights on the effect of HTLV-2 infection and protease inhibitors on the production of broadly neutralizing antibodies in PLH.
What is special about ritonavir-boosted protease inhibitors?
Is it only protease inhibiting activity and hence reduced viral maturation responsible for enhanced breadth of anti-HIV antibodies?
Is there any other virus that increases broadly antibodies in co-infected PLWH?
Is it known that B cell subset composition in PLWH is predictive of broadly neutralizing antibodies?
Author Response
This is very interesting and well conducted study, and a well written manuscript. The authors have investigated the impact of HTLV-2 on the presence of broadly neutralizing anti-HIV antibodies in HIV-infected individuals. They also looked into their effects of B cell subsets. It is sad that the researchers could not look into the frequencies of circulating TFH in the study participants. Nevertheless, the study provides very novel insights on the effect of HTLV-2 infection and protease inhibitors on the production of broadly neutralizing antibodies in PLH.
Comments 1: What is special about ritonavir-boosted protease inhibitors?
Response 1: Our interpretation is that the absence of protease activity due to antiretroviral treatment results in the production of immature virions, leading to a dispersed presentation of viral spikes and, consequently, altering their antigenicity and immunogenicity. Supporting this hypothesis is a very recent study showing that the incorporation of retroviral protease into VLPs enhances their ability to induce neutralizing responses.
Zhang P, Singh M, Becker VA, Croft J, Tsybovsky Y, Gopan V, et al. Inclusion of a retroviral protease enhances the immunogenicity of VLP-forming mRNA vaccines against HIV-1 or SARS-CoV-2 in mice. Sci Transl Med 2025; 17:eadt9576. doi: 10.1126/scitranslmed.adt9576.
We have added a sentence to include the findings of this study in the Discussion section.
Comments 2: Is it only protease inhibiting activity and hence reduced viral maturation responsible for enhanced breadth of anti-HIV antibodies?
Response 2: We appreciate this comment and fully acknowledge that additional factors may contribute to the induction of cross-neutralizing responses. Nevertheless, based on the data currently available, we believe that the most plausible explanation for the differences observed between the groups treated with and without b-PI relates primarily to the distinct morphology and antigenicity of the resulting virions.
However, we acknowledge that other factors cannot be excluded and have revised the Discussion accordingly.
Comments 3: Is there any other virus that increases broadly antibodies in co-infected PLWH?
Response 3: Indeed, certain viral coinfections can lead to broad activation of the immune system, including increased broadly reactive or nonspecific antibody production in people living with HIV (PLWH). The key mechanisms involve polyclonal B-cell activation, immune dysregulation due to chronic immune activation, and altered T-cell help.
Examples include herpesviruses such as Epstein-Barr virus (EBV), which infects B cells and induces polyclonal B-cell activation, and cytomegalovirus (CMV), which drives strong chronic immune activation, including T-cell and B-cell activation. CMV is associated with increased levels of immune activation, including nonspecific antibody production, and sometimes more broadly reactive antibodies, possibly due to bystander effects or epitope spreading. We did not assess antibody levels against herpesviruses such as CMV, HSV, or EBV. However, these viruses are detectable in >90% of former intravenous drug users. Given that both study groups consisted of former injecting drug users, we can reasonably assume comparable herpesvirus prevalence between groups. We acknowledge this limitation in the Discussion section.
Hepatitis C virus (HCV) is linked to immune dysregulation and mixed cryoglobulinemia (immune complex disease driven by B-cell expansion), while HBV enhances immune activation. These coinfections are reported in Table 1.
Comments 4: Is it known that B cell subset composition in PLWH is predictive of broadly neutralizing antibodies?
Response 4: Several scientific studies have demonstrated that the composition and characteristics of B cell subsets in people living with HIV (PLWH) are predictive of the development of broadly neutralizing antibodies (bNAbs). These findings are crucial for informing HIV vaccine strategies. We added these studies to clarify this issue:
A study published in Nature Immunology analyzed antibody heavy-chain repertoires in a large cohort of HIV-infected individuals. The researchers found that individuals who developed bNAbs exhibited distinct B cell repertoire features, including higher frequencies of B cells with long complementarity-determining region 3 (CDR-H3) loops and high somatic hypermutation rates. These features were not observed in individuals without bNAbs or in uninfected controls, suggesting that specific B cell repertoire characteristics are associated with the development of neutralizing antibody breadth.
Roskin KM, Jackson KJL, Lee JY, et al. Aberrant B Cell Repertoire Selection Associated with HIV Neutralizing Antibody Breadth. Nature Immunology 2020. doi: 10.1038/s41590-019-0581-0.
Another research published in eLife investigated B cell clonal evolution in HIV controllers—individuals who naturally control HIV infection without antiretroviral therapy. The study found that these individuals exhibited distinct B cell clonal evolution patterns associated with the development of neutralizing antibody breadth. This suggests that specific B cell subset dynamics contribute to the generation of bNAbs in PLWH.
Cizmeci D, Lofano G, Rossignol E, et al. Distinct Clonal Evolution of B Cells in HIV Controllers with Neutralizing Antibody Breadth. ELife 2021; 10: e62648. DOI: 10.7554/eLife.62648
Another study examined the frequency and phenotype of HIV envelope-specific B cells in patients with broadly cross-neutralizing antibodies. The researchers found that these patients had higher frequencies of envelope-specific B cells, particularly within the plasmablast subset, compared to those without such antibodies. This suggests that the presence and characteristics of specific B cell subsets are associated with the development of bNAbs.
Doria-Rose NA, Klein RM, Manion MM, et al. Frequency and phenotype of Human Immunodeficiency Virus Envelope-Specific B Cells from Patients with Broadly Cross-Neutralizing Antibodies. J Virol 2009; 83: 188-99. DOI: 10.1128/JVI.01583-08.
Round 2
Reviewer 1 Report
Comments and Suggestions for Authors
The authors have responded properly to the comments and the manuscript has been significantly improved and now may be publishable